# Robust exploration in linear quadratic reinforcement learning

**Jack Umenberger**
Department of Information Technology
Uppsala University, Sweden
jack.umenberger@it.uu.se

**Mina Ferizbegovic**
School of Electrical Engineering
and Computer Science
KTH, Sweden
minafe@kth.se

**Thomas B. Schön**
Department of Information Technology
Uppsala University, Sweden
thomas.schon@it.uu.se

**Håkan Hjalmarsson**
School of Electrical Engineering
and Computer Science
KTH, Sweden
hjalmars@kth.se

## Abstract

This paper concerns the problem of learning control policies for an unknown linear dynamical system to minimize a quadratic cost function. We present a method, based on convex optimization, that accomplishes this task *robustly*: i.e., we minimize the worst-case cost, accounting for system uncertainty given the observed data. The method balances exploitation and exploration, exciting the system in such a way so as to reduce uncertainty in the model parameters to which the worst-case cost is most sensitive. Numerical simulations and application to a hardware-in-the-loop servo-mechanism demonstrate the approach, with appreciable performance and robustness gains over alternative methods observed in both.

## 1 Introduction

Learning to make decisions in an uncertain and dynamic environment is a task of fundamental importance in a number of domains. Though it has been the subject of intense research activity since the formulation of the 'dual control problem' in the 1960s[17], the recent success of reinforcement learning (RL), particularly in games [33, 37], has inspired a resurgence in interest in the topic. Problems of this nature require decisions to be made with respect to two objectives. First, there is a goal to be achieved, typically quantified as a reward function to be maximized. Second, due to the inherent uncertainty there is a need to gather information about the environment, often referred to as 'learning' via 'exploration'. These two objectives are often competing, a fact known as the exploration/exploitation trade-off in RL, and the 'dual effect' (of decision) in control.

It is important to recognize that the second objective (exploration) is important only in so far as it facilitates the first (maximizing reward); there is no intrinsic value in reducing uncertainty. As a consequence, exploration should be targeted or application specific; it should *not* excite the system arbitrarily, but rather in such a way that the information gathered is useful for achieving the goal. Furthermore, in many real-world applications, it is desirable that exploration does not compromise the safe and reliable operation of the system.

This paper is concerned with control of uncertain linear dynamical systems, with the goal of maximizing (minimizing) rewards (costs) that are a quadratic function of states and actions; cf. §2 for a detailed problem formulation. We derive methods to synthesize control policies that balance the

exploration/exploitation tradeoff by performing robust, targeted exploration: *robust* in the sense that we optimize for worst-case performance given uncertainty in our knowledge of the system, and *targeted* in the sense that the policy excites the system so as to reduce uncertainty in such a way that specifically minimizes the worst-case cost. To this end, this paper makes the following specific contributions. We derive a high-probability bound on the spectral norm of the system parameter estimation error, in a form that is applicable to both robust control synthesis and design of targeted exploration; cf. §3. We also derive a convex approximation of the worst-case (w.r.t. parameter uncertainty) infinite-horizon linear quadratic regulator (LQR) problem; cf. §4.2. We then combine these two developments to present an approximate solution, via convex semidefinite programing (SDP), to the problem of minimizing the worst-case quadratic costs for an uncertain linear dynamical system; cf. §4. For brevity, we will refer to this as a 'robust reinforcement learning' (RRL) problem.

## 1.1 Related work

Inspired, perhaps in part, by the success of RL in games [33, 37], there has been a flurry of recent research activity in the analysis and design of RL methods for linear dynamical systems with quadratic rewards. Works such as [1, 23, 15] employ the so-called 'optimism in the face of uncertainty' (OFU) principle, which selects control actions assuming that the true system behaves as the 'best-case' model in the uncertain set. This leads to optimal regret but requires the solution of intractable non-convex optimization problems. Alternatively, the works of [35, 3, 4] employ Thompson sampling, which optimizes the control action for a system drawn randomly from the posterior distribution over the set of uncertain models, given data. The work of [30] eschews uncertainty quantification, and demonstrates that 'so-called' certainty equivalent control attains optimal regret. There has also been considerable interest in 'model-free' methods[39] for direct policy optimization [16, 29], as well partially model-free methods based on spectral filtering [22, 21]. Unlike the present paper, none of the works above consider robustness which is essential for implementation on physical systems. Robustness is studied in the so-called 'coarse-ID' family of methods, c.f. [12, 13, 11]. In [11], sample convexity bounds are derived for LQR with unknown linear dynamics. This approach is extended to adaptive LQR in [12], however, unlike the present paper, the policies are not optimized for exploration and exploitation jointly; exploration is effectively random. Also of relevance is the field of so-called 'safe RL' [19] in which one seeks to respect certain safety constraints during exploration and/or policy optimization [20, 2], as well as 'risk-sensitive RL', in which the search for a policy also considers the variance of the reward [32, 14]. Other works seek to incorporate notions of robustness commonly encountered in control theory, e.g. stability [34, 7, 10]. In closing, we mention that related problems of simultaneous learning and control have a long history in control theory, beginning with the study of 'dual control' [17, 18] in the 1960s. Many of these formulations relied on a dynamic programing (DP) solution and, as such, were applicable only in special cases [6, 8]. Nevertheless, these early efforts [9] established the importance of balancing 'probing' (exploration) with 'caution' (robustness). For subsequent developments from the field of control theory, cf. e.g. [24, 25, 5].

## 2 Problem statement

In this section we describe in detail the problem addressed in this paper. Notation is as follows: $A^\top$ denotes the transpose of a matrix $A$. $x_{1:n}$ is shorthand for the sequence $\{x_t\}_{t=1}^n$. $\lambda_{\max}(A)$ denotes the maximum eigenvalue of a matrix $A$. $\otimes$ denotes the Kronecker product. $\text{vec}(A)$ stacks the columns of $A$ to form a vector. $\mathbb{S}_+^n$ ($\mathbb{S}_{++}^n$) denotes the cone(s) of $n \times n$ symmetric positive semidefinite (definite) matrices. w.p. means 'with probability.' $\chi_n^2(p)$ denotes the value of the Chi-squared distribution with $n$ degrees of freedom and probability $p$. blkdiag is the block diagonal operator.

**Dynamics and cost function**   We are concerned with control of linear time-invariant systems

$$x_{t+1} = Ax_t + Bu_t + w_t, \quad w_t \sim \mathcal{N}\left(0, \sigma_w^2 I_{n_x}\right), \quad x_0 = 0, \tag{1}$$

where $x_t \in \mathbb{R}^{n_x}$, $u_t \in \mathbb{R}^{n_u}$ and $w_t \in \mathbb{R}^n$ denote the state (which is assumed to be observed directly, without noise), input and process noise, respectively, at time $t$. The objective is to design a feedback control policy $u_t = \phi(\{x_{1:t}, u_{1:t-1}\})$ so as to minimize the cost function $\sum_{t=i}^T c(x_t, u_t)$, where $c(x_t, u_t) = x_t^\top Q x_t + u_t^\top R u_t$ for user-specified positive semidefinite matrices $Q$ and $R$. When the parameters of the true system, denoted $\{A_{\text{tr}}, B_{\text{tr}}\}$, are known this is exactly the finite-horizon LQR problem, the optimal solution of which is well-known. We assume that $\{A_{\text{tr}}, B_{\text{tr}}\}$ are unknown.

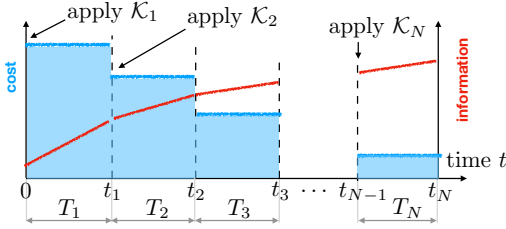

Figure 1: Cartoon depiction of the problem addressed in this paper. The goal is to design $N$ policies, $\{\mathcal{K}_i\}_{i=1}^N$, so as to minimize the worst-case cost (blue area) over the time horizon $[0, T]$; cf. §2.

**Modeling and data**  As $\{A_{\mathrm{tr}}, B_{\mathrm{tr}}\}$ are unknown, all knowledge about the true system dynamics must be inferred from observed data, $\mathcal{D}_n := \{x_t, u_t\}_{t=1}^n$. We assume that $\sigma_w$ is known, or has been estimated, and that we have access to initial data, denoted (with slight notational abuse) $\mathcal{D}_0$, obtained, e.g. during a preliminary experiment. For the model (1), parameter uncertainty can be quantified as:

**Proposition 2.1.** *Given observed data $\mathcal{D}_n$ from (1), and a uniform prior over the parameters $\theta = \mathrm{vec}\,([A\ B])$, i.e., $p(\theta) \propto 1$, the posterior distribution $p(\theta|\mathcal{D}_n)$ is given by $\mathcal{N}\,(\mu_\theta, \Sigma_\theta)$, where $\mu_\theta = \mathrm{vec}\left([\hat{A}\ \hat{B}]\right) = \arg\min_{\theta \in \mathbb{R}^{n_x^2 + n_x n_u}} \sum_{t=1}^{n-1} |x_{t+1} - [x_t^\top\ u_t^\top] \otimes I_{n_x}\theta|^2$, i.e., the ordinary least squares estimator, and $\Sigma_\theta^{-1} = \frac{1}{\sigma_w^2} \sum_{t=1}^{n-1} \begin{bmatrix} x_t \\ u_t \end{bmatrix} \begin{bmatrix} x_t \\ u_t \end{bmatrix}^\top \otimes I_{n_x}$.*

*Proof*: cf. §A.1.1. The uniform prior, $p(\theta) \propto 1$, sometimes called an *improper prior*, is used as an uninformative prior, signifying that we have no prior knowledge of $\theta$, i.e., all values are equally likely. Based on Proposition 2.1 we can define a high-probability credibility region by:

$$\Theta_e(\mathcal{D}_n) := \{\theta\ :\ (\theta - \mu_\theta)^\top \Sigma_\theta^{-1} (\theta - \mu_\theta) \leq c_\delta\}, \tag{2}$$

where $c_\delta = \chi^2_{n_x^2 + n_x n_u}(\delta)$ for $0 < \delta < 1$. Then, $\theta_{\mathrm{tr}} = \mathrm{vec}\,([A_{\mathrm{tr}}\ B_{\mathrm{tr}}]) \in \Theta_e(\mathcal{D}_n)$ w.p. $1 - \delta$.

**Policies**  Though not necessarily optimal, we will restrict our attention to static-gain policies of the form $u_t = Kx_t + \Sigma^{1/2}e_t$, where $e_t \sim \mathcal{N}\,(0, I)$ represent random excitations for the purpose of learning. Static-gain policies are popular in practice, due to simplicity of synthesis and implementation [12, 13, 11], and encompass many common control strategies, e.g., proportional-derivative (PD) control. A policy comprises $K \in \mathbb{R}^{p \times n}$ and $\Sigma \in \mathbb{S}_+^{n_u}$, and is denoted $\mathcal{K} = \{K, \Sigma\}$. Let $\{t_i\}_{i=0}^N \in \mathbb{N}$, with $0 = t_0 \leq t_1 \leq \ldots, \leq t_N = T$, partition the time horizon $T$ into $N$ intervals. The $i$th interval is of length $T_i := t_i - t_{i-1}$. We will then design $N$ policies, $\{\mathcal{K}_i\}_{i=1}^N$, such that $\mathcal{K}_i = \{K_i, \Sigma_i\}$ is deployed during the $i$th interval, $t \in [t_{i-1}, t_i]$. For convenience, we define the function $\mathcal{I} : \mathbb{R}_+ \mapsto \mathbb{N}$ given by $\mathcal{I}(t) := \arg\min_{i \in \mathbb{N}}\{i\ :\ t \leq t_j\}$, which maps time $t$ to the index $i = \mathcal{I}(t)$ of the policy to be deployed. We also make use of the notation $u_t = \mathcal{K}(x_t)$ as shorthand for $u_t = Kx_t + \Sigma^{1/2}e_t$.

**Worst-case dynamics**  We are now in a position to define the optimization problem that we wish to solve in this paper. In the absence of knowledge of the true dynamics, $\{A_{\mathrm{tr}}, B_{\mathrm{tr}}\}$, given initial data $\mathcal{D}_0$, we wish to find a sequence of policies $\{\mathcal{K}_i\}_{i=0}^N$ that minimize the expected cost $\sum_{t=1}^T c(x_t, u_t)$, assuming that, at time $t$, the system evolves according to the *worst-case* dynamics within the high-probability credibility region $\Theta_e(\mathcal{D}_t)$, i.e.,

$$\min_{\{\mathcal{K}_i\}_{i=1}^N} \mathbb{E}\left[\sum_{t=0}^T \sup_{\{A_t, B_t\} \in \Theta_e(\mathcal{D}_t)} c(x_t, u_t)\right],\ \text{s.t.}\ x_{t+1} = A_t x_t + B_t u_t + w_t,\ u_t = \mathcal{K}_{\mathcal{I}(t)}(x_t), \tag{3}$$

where the expectation is w.r.t. $w_t \sim \mathcal{N}\,(0, \sigma_w^2 I_{n_x})$ and $e_t \sim \mathcal{N}\,(0, I_{n_u})$. We choose to optimize for the worst-case dynamics so as to bound, with high probability, the cost of applying the policies to the unknown true system. In principle, problems such as (3) can be solved via *dynamic programing* (DP) [17]. However, such DP-based solutions require gridding to obtain finite state-action spaces, and are hence computationally intractable for systems of even modest dimension [6]; cf. also [31, §IV] for a discussion of RL methods for finite state-action spaces. In what follows, we will present an approximate solution to this problem, which we refer to as a 'robust reinforcement learning' (RRL) problem, that retains the continuous sate-action space formulation and is based on convex optimization. To facilitate such a solution, we require a refined means of quantifying system uncertainty, which we present next.

# 3 Modeling uncertainty for robust control

In this paper, we adopt a model-based approach to control, in which quantifying uncertainty in the estimates of the system dynamics is of central importance. From Proposition 2.1 the posterior distribution over parameters is Gaussian, which allows us to construct an 'ellipsoidal' credibility region $\Theta_e$, centered about the ordinary least squares estimates of the model parameters, as in (2).

To allow for an exact convex formulation of the control problem involving the *worst-case* dynamics, cf. §4.2, it is desirable to work with a credibility region that bounds uncertainty in terms of the spectral properties of the parameter error *matrix* $[\hat{A} - A_{\mathrm{tr}}, \ \hat{B} - B_{\mathrm{tr}}]$, where $\{\hat{A}, \hat{B}\}$ are the ordinary least squares estimates, i.e. $\mathrm{vec}\left([\hat{A}\,\hat{B}]\right) = \mu_\theta$, cf. Proposition 2.1. To this end, we will work with models of the form $\mathcal{M}(\mathcal{D}) := \{\hat{A}, \hat{B}, D\}$ where $D \in \mathbb{S}^{n_x+n_u}$ specifies the following region, in parameter space, centered about $\{\hat{A}, \ \hat{B}\}$:

$$\Theta_m(\mathcal{M}) := \{A, \ B \ : \ X^\top D X \preceq I, \ X = [\hat{A} - A, \ \hat{B} - B]^\top\}. \tag{4}$$

The following lemma, cf. §A.1.2 for proof, suggests a specific means of constructing $D$, so as to ensure that $\Theta_m$ defines a high-probability credibility region:

**Lemma 3.1.** *Given data $\mathcal{D}_n$ from* (1), *and* $0 < \delta < 1$, *let* $D = \frac{1}{\sigma_w^2 c_\delta} \sum_{t=1}^{n-1} \begin{bmatrix} x_t \\ u_t \end{bmatrix} \begin{bmatrix} x_t \\ u_t \end{bmatrix}^\top$, *with* $c_\delta = \chi^2_{n_x^2 + n_x n_u}(\delta)$. *Then* $[A_{\mathrm{tr}}, \ B_{\mathrm{tr}}] \in \Theta_m(\mathcal{M})$ *w.p.* $1 - \delta$.

For convenience, we will make use of the following shorthand notation: $\mathcal{M}(\mathcal{D}_{t_i}) = \{\hat{A}_i, \hat{B}_i, D_i\}$.

Credibility regions of the form (4), i.e. bounds on the spectral properties of the estimation error, have appeared in recent works on data-driven and adaptive control, cf. e.g., [11, Proposition 2.4] which makes use of results from high-dimensional statistics [40]. The construction in [11, Proposition 2.4] requires $\{x_{t+1}, x_t, u_t\}$ to be independent, and as such is not directly applicable to time series data, without subsampling to attain uncorrelated samples (though more complicated extensions to circumvent this limitation have been suggested [38]). Lemma 3.1 is directly applicable to correlated time series data, and provides a credibility region that is well suited to the RRL problem, cf. §4.3.

# 4 Convex approximation to robust reinforcement learning problem

Equipped with the high-probability bound on the spectral properties of the parameter estimation error presented in Lemma 3.1, we now proceed with the main contribution of this paper: a convex approximation to the 'robust reinforcement learning' (RRL) problem in (3).

## 4.1 Steady-state approximation of cost

In pursuit of a more tractable formulation, we first introduce the following approximation of (3),

$$\sum_{i=1}^{N} \sup_{\substack{\{A,B\}\in \\ \Theta_m(\mathcal{M}(\mathcal{D}_{t_i}))}} \mathbb{E}\left[ \sum_{t=t_{i-1}}^{t_i} c(x_t, u_t) \right], \ \text{s.t. } x_{t+1} = Ax_t + Bu_t + w_t, \ u_t = \mathcal{K}_i(x_t). \tag{5}$$

Observe that (5) has introduced two approximations to (3). First, in (5) we only update the 'worst-case' model at the beginning of each epoch, when we deploy a new policy, rather than at each time step as in (3). This introduces some conservatism, as model uncertainty will generally decrease as more data is collected, but results in a simpler control synthesis problem. Second, we select the worst-case model from the 'spectral' credibility region $\Theta_m$ as defined in (4), rather than the 'ellipsoidal' region $\Theta_e$ defined in (2). Again, this introduces some conservatism as $\Theta_e \subseteq \Theta_m$, cf. §A.1.2, but permits convex optimization of the worst-case cost, cf. §4.2. For convenience, we denote

$$J_\tau(x_1, \mathcal{K}, \Theta_m(\mathcal{M})) := \sup_{\substack{\{A,B\}\in \\ \Theta_m(\mathcal{M})}} \sum_{t=1}^{\tau} c(x_t, u_t), \ \text{s.t. } x_{t+1} = Ax_t + Bu_t + w_t, \ u_t = \mathcal{K}(x_t).$$

Next, we approximate the cost between epochs with the infinite-horizon cost, scaled appropriately for the epoch duration, i.e., between the $i-1$th and $i$th epoch we approximate the cost as

$$J_{T_i}(x_{t_i}, \mathcal{K}_i, \Theta_m(\mathcal{M}(\mathcal{D}_{t_{i-1}}))) \approx T_i \times \{J_\infty(\mathcal{K}_i, \Theta_m(\mathcal{M}(\mathcal{D}_{t_{i-1}}))) := \lim_{\tau \to \infty} \frac{1}{\tau} J_\tau(0, \mathcal{K}_i, \Theta_m(\mathcal{M}(\mathcal{D}_{t_{i-1}})))\}. \tag{6}$$

This approximation is accurate when the epoch duration $T_i$ is sufficiently long relative to the time required for the state to reach the stationary distribution. Substituting (6) into (5), the cost function that we seek to minimize becomes

$$\mathbb{E}\left[\sum_{i=1}^N T_i \times J_\infty\left(\mathcal{K}_i, \Theta_m(\mathcal{M}(\mathcal{D}_{t_{i-1}}))\right)\right]. \tag{7}$$

The expectation in (7) is w.r.t. to $w_t$ and $e_t$, as $\mathcal{D}_{t_i}$ depends on the random variables $x_{1:t_i}$ and $u_{1:t_i}$, which evolve according to the worst-case dynamics in (5).

## 4.2 Optimization of worst-case cost

The previous subsection introduced an approximation of our 'ideal' problem (3), based on the worst-case infinite horizon cost, cf. (7). In this subsection we present a convex approach to the optimization of $J_\infty(\mathcal{K}, \Theta_m(\mathcal{M}))$ w.r.t. $\mathcal{K}$, given $\mathcal{M}$. The infinite horizon cost can be expressed as

$$\lim_{\tau \to \infty} \frac{1}{\tau} \mathbb{E}\left[\sum_{t=1}^\tau x_t^\top Q x_t + u_t^\top R u_t\right] = \text{tr}\left(\begin{bmatrix} Q & 0 \\ 0 & R \end{bmatrix} \lim_{\tau \to \infty} \frac{1}{\tau} \sum_{t=1}^\tau \mathbb{E}\left[\begin{bmatrix} x_t \\ u_t \end{bmatrix} \begin{bmatrix} x_t \\ u_t \end{bmatrix}^\top\right]\right). \tag{8}$$

Under the feedback policy $\mathcal{K}$, the covariance appearing on the RHS of (8) can be expressed as

$$\mathbb{E}\left[\lim_{\tau \to \infty} \frac{1}{\tau} \sum_{t=1}^\tau \begin{bmatrix} x_t \\ Kx_t + \Sigma^{1/2}e_t \end{bmatrix} \begin{bmatrix} x_t \\ Kx_t + \Sigma^{1/2}e_t \end{bmatrix}^\top\right] = \begin{bmatrix} W & WK^\top \\ KW & KWK^\top + \Sigma \end{bmatrix}, \tag{9}$$

where $W = \mathbb{E}\left[x_t x_t^\top\right]$ denotes the stationary state covariance. For known $A$ and $B$, $W$ is given by the (minimum trace) solution to the Lyapunov inequality

$$W \succeq (A + BK)W(A + BK)^\top + B\Sigma B^\top + \sigma_w^2 I_{n_x}, \tag{10}$$

i.e., $\arg\min_W \text{tr } W$ s.t. (10). To optimize $J_\infty(\mathcal{K}, \Theta_m(\mathcal{M}))$ via convex optimization, there are two challenges to overcome: i. non-convexity of jointly searching for $K$ and $W$, satisfying (10) and minimizing (8), ii. computing $W$ for worst-case $\{A, B\} \in \Theta_m(\mathcal{M})$, rather than known $\{A, B\}$.

Let us begin with the first challenge: nonconvexity. To facilitate a convex formulation of the RRL problem (7) we write (10) as

$$W \succeq [A\ B] \begin{bmatrix} W & WK^\top \\ KW & KWK^\top + \Sigma \end{bmatrix} [A\ B]^\top + \sigma_w^2 I_{n_x}, \tag{11}$$

and introduce the change of variables $Z = WK^\top$ and $Y = KWK^\top + \Sigma$, collated in the variable $\Xi = \begin{bmatrix} W & Z \\ Z^\top & Y \end{bmatrix}$. With this change of variables, minimizing (8) subject to (11) is a convex program.

Now, we turn to the second challenge: computation of the stationary state covariance under the worst-case dynamics. As a sufficient condition, we require (11) to hold for all $\{A, B\} \in \Theta_m(\mathcal{M})$. In particular, we define the following approximation of $J_\infty(\mathcal{K}, \mathcal{M})$

$$\tilde{J}_\infty(\mathcal{K}, \mathcal{M}) := \min_{W \in \mathbb{S}_{++}^{n_x}} \text{tr}\left(\begin{bmatrix} Q & 0 \\ 0 & R \end{bmatrix} \begin{bmatrix} W & WK^\top \\ KW & KWK^\top + \Sigma \end{bmatrix}\right), \text{ s.t. (11) holds } \forall \{A, B\} \in \Theta_m(\mathcal{M}). \tag{12}$$

**Lemma 4.1.** *Consider the worst-case cost $J_\infty(\mathcal{K}, \mathcal{M})$, cf. (6), and the approximation $\tilde{J}_\infty(\mathcal{K}, \mathcal{M})$, cf. (12). $\tilde{J}_\infty(\mathcal{K}, \mathcal{M}) \geq J_\infty(\mathcal{K}, \mathcal{M})$.*

*Proof:* cf. §A.1.3. To optimize $\tilde{J}_\infty(\mathcal{K}, \mathcal{M})$, as defined in (12), we make use of the following result from [28]:

**Theorem 4.1.** *The data matrices $(\mathcal{A},\mathcal{B},\mathcal{C},\mathcal{P},\mathcal{F},\mathcal{G},\mathcal{H})$ satisfy, for all $X$ with $I - X^\top \mathcal{P} X \succeq 0$, the robust fractional quadratic matrix inequality*

$$\begin{bmatrix} \mathcal{H} & \mathcal{F}+\mathcal{G}X \\ (\mathcal{F}+\mathcal{G}X)^\top & \mathcal{C}+X^\top\mathcal{B}+\mathcal{B}^\top X+X^\top\mathcal{A}X \end{bmatrix} \succeq 0, \text{ iff } \begin{bmatrix} \mathcal{H} & \mathcal{F} & \mathcal{G} \\ \mathcal{F}^\top & \mathcal{C}-\lambda I & \mathcal{B}^\top \\ \mathcal{G}^\top & \mathcal{B} & \mathcal{A}+\lambda\mathcal{P} \end{bmatrix} \succeq 0,$$

(13)

*for some $\lambda \geq 0$.*

To put (11) in a form to which Theorem 4.1 is applicable, we make use of of the nominal parameters $\hat{A}$ and $\hat{B}$. With $X$ defined as in (4), such that $[A\ B] = [\hat{A}\ \hat{B}] - X'$, we can express (11) as

$$\Psi := W - [\hat{A}\ \hat{B}]\Xi[\hat{A}\ \hat{B}]^\top + X^\top\Xi[\hat{A}\ \hat{B}]^\top + [\hat{A}\ \hat{B}]\Xi X - X^\top\Xi X \succeq \sigma_w^2 I_{n_x} \iff \begin{bmatrix} I & \sigma_w I \\ \sigma_w I & \Psi \end{bmatrix} \succeq 0,$$

where the 'iff' follows from the Schur complement. Given this equivalent representation, by Theorem 4.1, (11) holds for all $X^\top D\mathcal{M} \preceq I$ (i.e. all $\{A,B\}\in\Theta_m(\mathcal{M})$) iff

$$S(\lambda,\Xi,\hat{A},\hat{B},D) := \begin{bmatrix} I & \sigma_w I & 0 \\ \sigma_w I & W-[\hat{A}\ \hat{B}]\Xi[\hat{A}\ \hat{B}]^\top-\lambda I & [\hat{A}\ \hat{B}]\Xi^\top \\ 0 & \Xi[\hat{A}\ \hat{B}]^\top & \lambda D - \Xi \end{bmatrix} \succeq 0,$$

(14)

which is simply (13) with the substitutions $\mathcal{A}=-\Xi$, $\mathcal{B}=\Xi[\hat{A}\ \hat{B}]^\top$, $\mathcal{C}=W-[\hat{A}\ \hat{B}]\Xi[\hat{A}\ \hat{B}]^\top$, $\mathcal{F}=\sigma_w I$, $\mathcal{G}=0$, and $\mathcal{P}=D$. We now have the following result, cf. §A.1.4 for proof.

**Theorem 4.2.** *The solution to $\min_\mathcal{K}\ \tilde{J}_\infty(\mathcal{K},\Theta_m(\mathcal{M}))$, cf. (12), is given by the SDP:*

$$\min_{\lambda,\Xi} \text{ tr}\left(\texttt{blkdiag}(Q,R)\Xi\right), \text{ s.t. } S(\lambda,\Xi,\hat{A},\hat{B},D)\succeq 0,\ \lambda\geq 0,$$

(15)

*with the optimal policy given by $\mathcal{K}=\{Z^\top W^{-1},\ Y-Z^\top W^{-1}Z\}$.*

Note that as $\min_\mathcal{K}\ \tilde{J}_\infty(\mathcal{K},\Theta_m(\mathcal{M}))$ is purely an 'exploitation' problem $\Sigma\to 0$ in the above SDP; in general, $\Sigma\neq 0$ in the RRL setting (i.e. (7)) where exploration is beneficial.

### 4.3 Approximate uncertainty propagation

Let us now return to the RRL problem (7). Given a model $\mathcal{M}$, §4.2 furnished us with a convex method to minimize the worst-case cost. However, at time $t=0$, we have access only to data $\mathcal{D}_0$, and therefore, only $\mathcal{M}(\mathcal{D}_0)$. To optimize (7) we need to approximate the models $\{\mathcal{M}(\mathcal{D}_{t_i})\}_{i=1}^{N-1}$ based on the future data, $\{\mathcal{D}_{t_i}\}_{i=1}^{N-1}$, that we *expect* to see. To this end, we denote the approximate model, at time $t=t_j$ given data $\mathcal{D}_{t_i}$, by $\tilde{\mathcal{M}}_j(\mathcal{D}_{t_i}) := \{\tilde{A}_{j|i},\tilde{B}_{j|i},\tilde{D}_{j|i}\}\approx\mathbb{E}\left[\mathcal{M}(\mathcal{D}_{t_j})|\mathcal{D}_{t_i}\right]$. We now describe specific choices for $\tilde{A}_{j|i}$, $\tilde{B}_{j|i}$, and $\tilde{D}_{j|i}$, beginning with the latter.

Recall that the uncertainty matrix $D$ at the $i$th epoch is denoted $D_i$. The uncertainty matrix at the $i+1$th epoch is then given by $D_{i+1}=D_i+\frac{1}{\sigma_w^2 c_\delta}\sum_{t=t_i}^{t_{i+1}}\begin{bmatrix}x_t\\u_t\end{bmatrix}\begin{bmatrix}x_t\\u_t\end{bmatrix}^\top$. We approximate the empirical covariance matrix in this expression with the worst-case state covariance $W_i$ as follows:

$$\mathbb{E}\left[\sum_{t=t_i}^{t_{i+1}}\begin{bmatrix}x_t\\u_t\end{bmatrix}\begin{bmatrix}x_t\\u_t\end{bmatrix}^\top\right]\approx T_{i+1}\begin{bmatrix}W_i & W_iK_i^\top\\K_i^\top W_i & K_iW_iK_i^\top+\Sigma_i\end{bmatrix}=T_{i+1}\Xi_i.$$

(16)

This approximation makes use of the same calculation appearing in (9). The equality makes use of the change of variables introduced in §4.2. Note that in proof of Theorem 4.2, cf. §A.1.4, it was shown that $\Xi=\begin{bmatrix}W & WK^\top\\K^\top W & KWK^\top+\Sigma\end{bmatrix}$, when $\Xi$ is the solution of (15).

Next, we turn our attention to approximating the effect of future data on the nominal parameter estimates $\{\hat{A},\hat{B}\}$. Updating these (ordinary least squares) estimates based on the expected value of future observations involves difficult integrals that must be approximated numerically [27, §5]. To preserve convexity in our formulation, we approximate future nominal parameter estimates with the current estimates, i.e., given data $\mathcal{D}_{t_i}$ we set $\tilde{A}_{j|i}=\hat{A}_i$ and $\tilde{B}_{j|i}=\hat{B}_i$. To summarize, our approximate model at epoch $j$ is given by $\tilde{\mathcal{M}}_j(\mathcal{D}_{t_i})=\{\hat{A}_i,\hat{B}_i,D_i+\frac{1}{\sigma_w^2 c_\delta}\sum_{k=i+1}^{j}T_{k+1}\Xi_k\}$.

### 4.4 Final convex program and receding horizon application

We are now in a position to present a convex approximation to our original problem (3). By substituting $\tilde{J}_\infty(\cdot, \cdot)$ for $J_\infty(\cdot, \cdot)$, and $\tilde{\mathcal{M}}_i(\mathcal{D}_0)$ for $\mathcal{M}(\mathcal{D}_{t_i})$ in (7), we attain the cost function: $\sum_{i=1}^N T_i \times \tilde{J}_\infty\left(\mathcal{K}_i, \Theta_m(\tilde{\mathcal{M}}_{i-1}(\mathcal{D}_0))\right)$. Consider the $i$th term in this sum, which can be optimized via the SDP (15), with $D = \tilde{D}_{i-1|0}$. Notice two important facts: 1. for fixed multiplier $\lambda$, the uncertainty $\tilde{D}_{i-1|0}$ enters *linearly* in the constraint $S(\cdot) \succeq 0$, cf. (15); 2. $\tilde{D}_{i-1|0}$ is *linear* in the decision variables $\{\Xi_k\}_{k=1}^{i-1}$, cf. end of §4.3. Therefore, the constraint $S(\cdot) \succeq 0$ remains linear in the decision variables, which means that the cost function derived by substituting $\tilde{\mathcal{M}}_i(\mathcal{D}_0)$ into (7) can be optimized as an SDP, cf. (17) below.

Hitherto, we have considered the problem of minimizing the expected cost over time horizon $T$ given initial data $\mathcal{D}_0$. In practical applications, we employ a *receding horizon* strategy, i.e., at the $i$th epoch, given data $\mathcal{D}_{t_{i-1}}$, we find a sequence of policies $\{\mathcal{K}_j\}_{j=i}^{i+h}$ that minimize the approximate $h$-step-ahead expected cost

$$\hat{J}(i, h, \{\mathcal{K}_j\}_{j=i}^{i+h}, \mathcal{D}_{t_{i-1}}) := T_i \tilde{J}_\infty(\mathcal{K}_i, \Theta_m(\mathcal{M}(\mathcal{D}_{t_{i-1}}))) + \sum_{j=i+1}^{i+h} T_j \tilde{J}_\infty(\mathcal{K}_j, \Theta_m(\tilde{\mathcal{M}}(\mathcal{D}_{t_{i-1}}))),$$

and then apply $\mathcal{K}_i$ during the $i$th epoch. At the beginning of the $i + 1$th epoch, we repeat the process; cf. Algorithm 1. The problem $\min_{\{\mathcal{K}_j\}_{j=i}^N} \hat{J}(i, h, \{\mathcal{K}_j\}_{j=i}^{i+h}, \mathcal{D}_{t_{i-1}})$ can be solved as the SDP:

$$\min_{\lambda_i \geq 0, \{\Xi_j\}_{j=i}^{i+h}} \sum_{j=i}^{i+h} \mathrm{tr}\left(\mathtt{blkdiag}(Q, R)\, \Xi_j\right), \quad \text{s.t. } S(\lambda_i, \Xi_i, \hat{A}_i, \hat{B}_i, D_i) \succeq 0, \ \Xi_j \succeq 0\, \forall j, \quad \text{(17a)}$$

$$S\left(\lambda_j, \Xi_k, \hat{A}_i, \hat{B}_i, D_i + \frac{1}{\sigma_w^2 c_\delta} \sum_{k=i+1}^j T_{k+1}\Xi_k\right) \succeq 0 \text{ for } j = i+1, \ldots, i+h. \quad \text{(17b)}$$

**Selecting multipliers** For optimization of $\tilde{J}_\infty(\mathcal{K}, \mathcal{M})$ given a model $\mathcal{M}$, i.e., (15), the simultaneous search for the policy $\mathcal{K}$ and multiplier $\lambda$ is convex, as $D$ is fixed. However, in the RRL setting, '$D$' is a function of the decision variables $\Xi_i$, cf. (17b), and so the multipliers $\{\lambda_j\}_{j=i+1}^{i+h} \in \mathbb{R}_+^{h-1}$ must be specified in advance. We propose the following method of selecting the multipliers: given $\mathcal{D}_{t_{i-1}}$, solve $\bar{\mathcal{K}} = \arg\min_\mathcal{K} \tilde{J}_\infty(\mathcal{K}, \Theta_m(\mathcal{M}(\mathcal{D}_{t_{i-1}})))$ via the SDP (15). Then, compute the cost $\hat{J}(i, h, \{\bar{\mathcal{K}}\}_{j=i}^{i+h}, \mathcal{D}_{t_{i-1}})$ by solving (17), but with the policies fixed to $\bar{\mathcal{K}}$, and the multipliers $\{\lambda_j\}_{j=i}^{i+h} \in \mathbb{R}_+^h$ as free decision variables. In other words, approximate the worst-case cost of deploying the $\bar{\mathcal{K}}$, $h$ epochs into the future. Then, use the multipliers found during the calculation of this cost for control policy synthesis at the $i$th epoch.

**Computational complexity** The proposed method can be implemented via semidefinite programming (SDP) for which computational complexity is well-understood. In particular, the cost of solving the SDP (15) scales as $\mathcal{O}(\max\{m^3, \ mn^3, m^2 n^2\})$ [26], where $m = (1/2)n_x(n_x + 1) + (1/2)n_u(n_u + 1) + n_x n_u + 1$ denotes the dimensionality of the decision variables, and $n = 3n_x + n_u$ is the dimensionality of the LMI $S \succeq 0$. The cost of solving the SDP (17) is then given, approximately, by the cost of (15) multiplied by the horizon $h$.

## 5 Experimental results

**Numerical simulations** In this section, we consider the RRL problem with parameters

$$A_{\mathrm{tr}} = \begin{bmatrix} 1.1 & 0.5 & 0 \\ 0 & 0.9 & 0.1 \\ 0 & -0.2 & 0.8 \end{bmatrix}, \ B_{\mathrm{tr}} = \begin{bmatrix} 0 & 1 \\ 0.1 & 0 \\ 0 & 2 \end{bmatrix}, \ Q = I, \ R = \mathtt{blkdiag}(0.1, 1), \ \sigma_w = 0.5.$$

We partition the time horizon $T = 10^3$ into $N = 10$ equally spaced intervals, each of length $T_i = 100$. For robustness, we set $\delta = 0.05$. Each experimental trial consists of the following procedure. Initial

---

**Algorithm 1** Receding horizon application to true system

---

1: **Input:** initial data $\mathcal{D}_0$, confidence $\delta$, LQR cost matrices $Q$ and $R$, epochs $\{t_i\}_{i=1}^N$.
2: **for** $i = 1 : N$ **do**
3:     Compute/update nominal model $\mathcal{M}(\mathcal{D}_{t_{i-1}})$.
4:     Solve convex program (17).
5:     Recover policy $\mathcal{K}_i$: $K_i = Z_i^\top W_i^{-1}$ and $\Sigma_i = Y_i - Z_i^\top W_i^{-1} Z_i$.
6:     Apply policy to true system for $t_{i-1} < t \leq t_i$, which evolves according to (1) with $u_t = K_i x_t + \Sigma_i^{1/2} e_t$.
7:     Form $\mathcal{D}_{t_i} = \mathcal{D}_{t_{i-1}} \cup \{x_{t_{i-1}:t_i}, u_{t_{i-1}:t_i}\}$ based on newly observed data.
8: **end for**

---

data $\mathcal{D}_0$ is obtained by driving the system forward 6 time steps, excited by $\tilde{u}_t \sim \mathcal{N}(0, I)$. This open-loop experiment is repeated 100 times, such that $\mathcal{D}_0 = \{\tilde{x}_{1:6}^i, \tilde{u}_{1:6}\}_{i=1}^{100}$. We then apply three methods: i. `rrl` - the method proposed in §4.4, with look-ahead horizon $h = 10$; ii. `nom` - applying the 'nominal' robust policy $\mathcal{K}_i = \arg\min_\mathcal{K} \tilde{J}_\infty(\mathcal{K}, \Theta_m(\mathcal{M}(\mathcal{D}_{t_i})))$, i.e., a pure greedy exploitation policy, with no explicit exploration; iii. `greedy` - first obtaining a nominal robustly stabilizing policy as with `nom`, but then optimizing (i.e., increasing, if possible) the exploration variance $\Sigma$ until the `greedy` policy and the `rrl` policy have the same theoretical worst-case cost at the current epoch. This is a greedy exploration policy; iv. `ts` - Thompson sampling [4]; v. `rbst` - the robust adaptive-control synthesis method proposed in [13]. We perform 100 of these trials and plot the results in Figure 2.

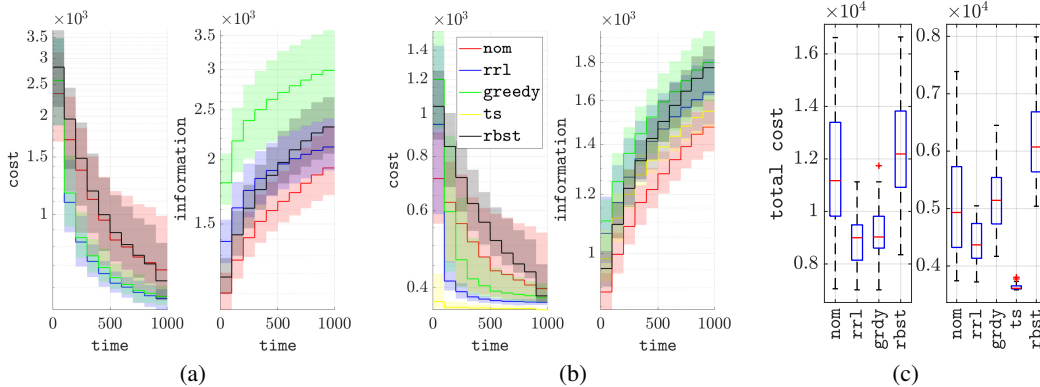

Figure 2: Results for the experiments described in §5. **(a)** cost and 'information' (a scalar measure of uncertainty defined in §5) when the system evolves according to the worst-case dynamics. The trace denotes the median, and the shaded region spans from the 10th to the 90th percentile. **(b)** cost and information when policies are applied to the true system. **(c)** sum of costs (over all time steps) for the worst-case dynamics (left) and the true system (right). Note that `greedy` is abbreviated as `grdy`.

In Figure 2(a) we plot the cost at each epoch when the system evolves according to the worst-case dynamics. we also plot the *information*, defined as $1/\lambda_{\max}(D_i^{-1})$, at the $i$th epoch, which is the (inverse) of the 2-norm of parameter error, cf. (4). This is a scalar measure of uncertainty: the larger the information, the more certain our estimate of the system (in an absolute sense). `ts` is omitted from these results as the closed-loop behavior diverges (i.e. attains infinite worst-case cost) in 96% of the trials conducted. In Figure 2(b) we plot the cost, and information, at each epoch when the policies are applied to the true system. Figure 2(c) plots the total cost (sum of costs over all epochs). We make the following observations. Concerning worst-case performance, `nom` attains the lowest cost at the initial epoch, as it does no explicit exploration. However, methods that incorporate exploration achieve better performance at subsequent epochs (and in terms of total cost) due to greater reduction in uncertainty. Of these methods, the proposed `rrl` performs best, optimally balancing exploration with exploitation; we emphasize that this balance of exploration/exploitation occurs *automatically*. Furthermore, observe that `greedy` actually achieves higher *information* (lower absolute uncertainty) relative to `rrl`, yet attains higher cost. This suggests that `rrl` is reducing the

uncertainty in a structured way, targeting uncertainty reduction in the parameters that 'matter most for control'. Results on the true system are qualitatively similar, with `rrl` attaining better performance than all other methods except `ts`. We note that Thompson sampling performs well when the policy happens to stabilize the system; however, as stability is not a consideration during `ts` synthesis, this cannot be guaranteed.

**Hardware-in-the-loop experiment**   In this section, we consider the RRL problem for a hardware-in-the-loop simulation comprised of the interconnection of a physical servo mechanism (Quanser QUBE 2) and a synthetic (simulated) LTI dynamical system. Control of servomechanisms is a ubiquitous task in practice (e.g. robotics); furthermore, the planar servo can be modeled reasonably well (globally) as a linear system. As such, this setup represents a good compromise between the complexities of a real world system (backlash, friction, unmodeled dynamics, disturbances, etc) and a system that approximately satisfies the assumptions of the method; cf. Appendix A.2 for full details of the experimental setup. An experimental trial consisted of the following procedure. Initial data was obtained by simulating the system for 0.5 seconds, under closed-loop feedback control (cf. Appendix A.2) with data sampled at 500Hz, to give 250 initial data points. We then applied methods `rrl` (with horizon $h = 5$) and `greedy` as described in §5. The total control horizon was $T = 1250$ (2.5 seconds at 500Hz) and was divided into $N = 5$ intervals, each of duration 0.5 seconds. We performed 5 of these experimental trials and plot the results in Figure 3. In Figure 3(b) and (d) we plot the total cost (the sum of the costs at each epoch), and the cost at each epoch, respectively, for each method, and observe significantly better performance from `rrl` in both cases. Additional plots decomposing the cost into that associated with the physical and synthetic system are available in Appendix A.2. We also applied `ts`, and observed that the resulting policy was unable to stabilize the system; cf. Figure 3(c) which demonstrates divergence of the angular position of the servomotor under `ts`.

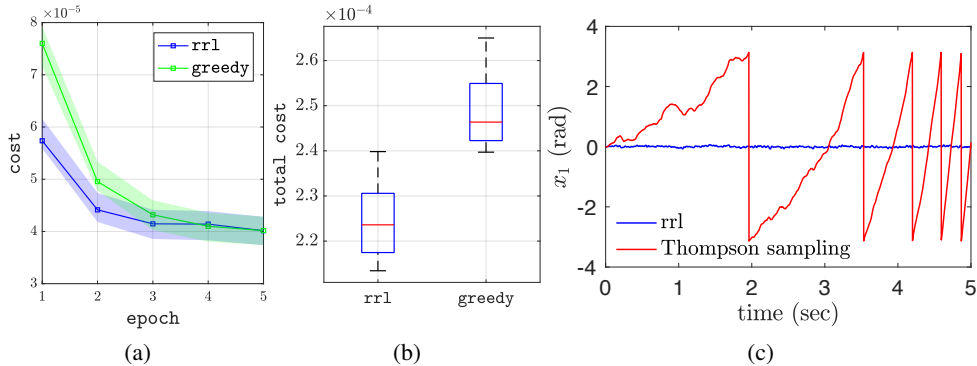

(a)  (b)  (c)

Figure 3: Results for the hardware-in-the-loop experiment in §5. **(a)** median costs at each epoch; the shaded region covers the best/worst costs at each epoch. **(b)** total costs (sum of costs at each epoch). **(c)** angular position of the servo motor ($x_1$) under feedback control with the proposed method and Thompson sampling; the latter results in divergence (uncontrolled revolutions of the servo motor).

## 6   Conclusion

We have presented an algorithm for robust, targeted exploration in RL for linear systems with quadratic rewards. Policies are robust in the sense that stability of the closed loop system is guaranteed, with high probability, during learning, and targeted, in the sense that uncertainty is reduced so as to improve performance of the controller on the specific task at hand. Roughly speaking, the policy prioritizes uncertainty reduction in the parameters that 'matter most for control'. The search for a policy is formulated as a convex program; solving to global optimality then automatically gives the optimal tradeoff between exploration and exploitation, in the worst-case setting.

**Acknowledgments**

This research was financially supported by the project *NewLEADS - New Directions in Learning Dynamical Systems* (contract number: 621-2016-06079), funded by the Swedish Research Council

and by the project *ASSEMBLE* (contract number: RIT15-0012), funded by the Swedish Foundation for Strategic Research (SSF).

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
