[Supplementary Material]

# A  Supplementary material

## A.1  Proofs

### A.1.1  Proof of Proposition 2.1

*Proof.* Observe that the dynamics in (1) can be rewritten as

$$x_{t+1} = Ax_t + Bu_t + w_t = x_t^\top \otimes I_{n_x} \mathrm{vec}\,(A) + u_t^\top \otimes I_{n_x} \mathrm{vec}\,(B) + w_t$$

$$= \underbrace{[x_t^\top\ u_t^\top] \otimes I_{n_x}}_{\Phi_t} \mathrm{vec}\,([A\ B]) + w_t$$

$$= \Phi_t \theta + w_t.$$

By Bayes' rule, the posterior distribution over parameters can be written as

$$p(\theta|\mathcal{D}_n) = \frac{1}{p(\mathcal{D}_n)} p(\mathcal{D}_n|\theta) p(\theta) \propto p(\mathcal{D}_n|\theta), \tag{18}$$

where proportionality follows from the assumption of a uniform prior, $p(\theta) \propto 1$. As $w_t \sim \mathcal{N}\left(0, \sigma_w^2 I\right)$ the likelihood can be expressed as

$$p(\mathcal{D}_n|\theta) = \prod_{t=1}^{n-1} p(x_{t+1}|x_t, u_t) \propto \exp\left(-\frac{1}{2\sigma_w^2}\sum_{t=1}^{n-1}|x_{t+1} - \Phi_t\theta|^2\right) =$$

$$\exp\left(-\frac{1}{2\sigma_w^2}\sum_{t=1}^{n-1} x_{t+1}^\top x_{t+1} - 2x_{t+1}^\top\Phi_t\theta + \theta'\Phi_t^\top\Phi_t\theta\right) \propto \exp\left(-\frac{1}{2}|\theta - \mu_\theta|_{\Sigma_\theta^{-1}}\right)$$

which has the norm of the Gaussian distribution $\mathcal{N}\,(\mu_\theta, \Sigma_\theta)$. From (18), we know that the posterior is proportional to the likelihood; therefore the posterior is given by $\mathcal{N}\,(\mu_\theta, \Sigma_\theta)$.

□

### A.1.2  Proof of Lemma 3.1

*Proof.* As $\theta = \mathrm{vec}\,([A\ B])$ and $\mu_\theta = \mathrm{vec}\left([\hat{A}\ \hat{B}]\right)$ we have $\theta - \mu_\theta = \mathrm{vec}\,(X^\top)$. Substituting this representation of $\theta - \mu_\theta$ into (2) we have, w.p. $1 - \delta$,

$$1 \geq \mathrm{vec}\left(X^\top\right)^\top \left(\frac{1}{\sigma_w^2 c_\delta}\sum_{t=1}^{n-1}\begin{bmatrix} x_t \\ u_t \end{bmatrix}\begin{bmatrix} x_t \\ u_t \end{bmatrix} \otimes I_{n_x}\right) \mathrm{vec}\left(X^\top\right) \tag{19a}$$

$$= \mathrm{tr}\left(XX^\top D\right) \tag{19b}$$

$$= \mathrm{tr}\left(X^\top D X\right) \tag{19c}$$

$$\geq \lambda_{\max}\left(X^\top D X\right), \tag{19d}$$

where (19a) is attained by dividing (2) by $c_\delta = \chi^2_{n_x^2+n_x n_u}(\delta)$; (19b) follows by combining the matrix identities

$$\mathrm{tr}\,A^\top B = \mathrm{vec}\,(A)^\top \mathrm{vec}\,(B) \tag{20}$$

c.f., [36, Equation 521], and

$$\mathrm{vec}\,(CEF) = (F \otimes C)\mathrm{vec}\,(E) \tag{21}$$

c.f., [36, Equation 520] to get

$$\mathrm{tr}\,XX^\top D = \mathrm{vec}\left(X^\top\right)^\top (D \otimes I)\,\mathrm{vec}\left(X^\top\right), \tag{22}$$

by choosing $A = X^\top$, $B = X^\top D$, $C = I$, $E = X^\top$ and $F = D^\top = D$; (19c) is simply the cyclic trace property; (19d) follows from the fact that the Frobenius norm upper bounds the spectral norm (2-norm) of a matrix. As $\lambda_{\max}\left(X^\top D X\right) \leq 1 \implies X^\top D X \preceq I$, this completes this proof. □

### A.1.3  Proof of Lemma 4.1

*Proof.* The worst-case cost $J_\infty(\mathcal{K}, \mathcal{M})$ is given by

$$\min_{W \in \mathbb{S}_{++}^{n_x}} \mathrm{tr}\left(\begin{bmatrix} Q & 0 \\ 0 & R \end{bmatrix}\begin{bmatrix} W & WK^\top \\ KW & KWK^\top + \Sigma \end{bmatrix}\right), \text{ s.t. (11) holds for } A = A_{wc},\ B = B_{wc}, \tag{23}$$

where $A_{wc}$ and $B_{wc}$ are the 'worst-case' $A$ and $B$, respectively, within $\Theta_m(\mathcal{M})$, as defined in (6).

The approximate cost $\tilde{J}_\infty(\mathcal{K}, \mathcal{M})$ is given by

$$\min_{W \in \mathbb{S}^{n_x}_{++}} \operatorname{tr}\left(\begin{bmatrix} Q & 0 \\ 0 & R \end{bmatrix}\begin{bmatrix} W & WK^\top \\ KW & KWK^\top + \Sigma \end{bmatrix}\right), \text{ s.t. (11) holds } \forall \{A, B\} \in \Theta_m(\mathcal{M}). \tag{24}$$

As the feasible set in (24) is a subset of the feasible set in (23), the cost of (24) cannot be less than that of (23). Therefore, $\tilde{J}_\infty(\mathcal{K}, \mathcal{M}) \geq J_\infty(\mathcal{K}, \mathcal{M})$.

$\square$

### A.1.4   Proof of Theorem 4.2

*Proof.* As Theorem 4.1 is non-conservative (i.e. if and only if), $\min_\mathcal{K} \tilde{J}_\infty(\mathcal{K}, \Theta_m(\mathcal{M}))$ is equivalent to solving

$$\min_{\lambda, W \succeq 0, K, \Sigma \succeq 0} \operatorname{tr}\left(\texttt{blkdiag}(Q, R)\bar{\Sigma}\right), \text{ s.t. } S(\lambda, \bar{\Sigma}, \hat{A}, \hat{B}, D) \succeq 0, \ \lambda \geq 0 \tag{25}$$

where

$$\bar{\Sigma} := \begin{bmatrix} W & WK^\top \\ KW & KWK^\top + \Sigma \end{bmatrix}.$$

When we solve the convex SDP (15) in Theorem 4.2, we solve with $\Xi \succeq 0$, as a free variable, instead of $\bar{\Sigma}$, i.e., we ignore the structural constraints implicit in $\bar{\Sigma}$. As we remove constraints from the problem, the SDP (15) has a solution that is at least as good as the solution to (25) (which in the optimal solution).

However, as we enforce $\Xi \succeq 0$, one can recover a feasible policy $K = Z^\top W^{-1}$ and $\Sigma = Y - Z^\top W^{-1} Z = Y - KWK^\top$, as the Schur complement implies

$$\Xi \succeq 0 \iff Y \succeq Z^\top W^{-1} Z \iff \Sigma := Y - KWK^\top \succeq 0. \tag{26}$$

Therefore, as the policy from the SDP (15) in Theorem 4.2 is: i) at least as good as the optimal policy, and ii) feasible, it must be equivalent to the optimal policy.

$\square$

### A.2   Description of hardware in the loop experiment

For the hardware-in-the-loop experiment described in §5, we consider a system comprised of two subsystems: i. a Quanser QUBE-Servo 2 physical (i.e. real-world) servomechanism, cf. Figure 4, and ii. a synthetic (i.e. simulated) LTI system of the form (1), with parameters

$$A_{syn} = \begin{bmatrix} 0.95 & 0.5 & 0 & 0 & 0 \\ 0 & 0.95 & 0.5 & 0 & 0 \\ 0 & 0 & 0.95 & 0 & 1 \\ 0 & 0 & 0 & -0.9 & 0.5 \\ 0 & 0 & 0 & 0.8 & -0.9 \end{bmatrix}, \quad B_{syn} = \begin{bmatrix} 0 \\ 0 \\ 0 \\ 0 \\ 1 \end{bmatrix}.$$

For the purpose of implementation in `MATLAB Simulink`, we set $C_{syn} = I_{5x5}$ and $D_{syn} = 0_{5x6}$ so as to output the full state $x_t$. The two subsystems are interconnected as depicted in the `Simulink` block diagram shown in Figure 5. The coupling between these two systems, cf. Figure 5, is $C_{coup} = \begin{bmatrix} 0.1 & 0 & 0 & 0 & 0 \end{bmatrix}$. Data was sampled from the physical system at 500 Hz, i.e., a sampling time of $T_s = 0.002$, and the position (measured directly via an encoder) was passed through a high-pass filter to obtain velocity estimates. Band-limited white noise (of unit power) was added to all states of the system, as shown in Figure 5. The gain for each 'noise channel' was set to $\sqrt{T_s} \times 10^{-3}$.

The experiment consisted of five trials, each comprising the following procedure. Initial data $\mathcal{D}_0$ was generated by simulating the system for 0.5 seconds, i.e. 250 samples at 500 Hz, under feedback control with the policy $\mathcal{K} = \{K, \Sigma\}$ given by

$$K = \begin{bmatrix} 2.1847 & 0.7384 & 0.0756 & 0.0625 & 0.0355 & -0.0087 & 0.0217 \\ -0.0062 & 0.0006 & 0.0789 & 0.3477 & 0.6417 & 1.7401 & -0.9099 \end{bmatrix}, \quad \Sigma = \sqrt{T_s} \times 10^{-3} I_{2 \times 2}.$$

We then applied the methods `rrl` and `greedy`, as defined in §5. The matrices specifying the cost function were given by $Q = \operatorname{diag}(1, 0.1, 0.1, 0.1, 0.1, 10, 0.1)$ and $R = \operatorname{diag}(0.1, 0.1)$. The total time horizon was $T = 1250$, i.e. 2.5 seconds at 500 Hz, which was divided into $N = 5$ equal intervals.

In Figure 6 we decompose the total cost plotted in Figure **??**(d) into the costs associated with the physical system and the synthetic (simulated) system.

Figure 4: The Quanser QUBE-Servo 2. Photo: www.quanser.com/products/qube-servo-2.

Figure 5: `Simulink` block diagram showing the interconnection of the physical system (Quarc) and the synthetic (simulated) system.

Figure 6: The (median) cost of `rrl` and `greedy` controllers on the synthetic system and the physical system. The shaded region covers the best and worst costs at each epoch.