[Reviews · NeurIPS 2019]

Reviewer 1



The paper is very well written and organized and its contributions are quite original as it proposes a novel coarse-ID method for robust model-based reinforcement learning in which both exploration AND exploitation are optimized jointly (which was not the case in previous similar works). The method proposed to solve the robust Reinforcement Learning problem is all the more original as it does not rely on Stochastic Dynamic Programming, but rather on Semidefinite Programming. Concerning clarity, the only element that is not clear for me is related to equation (1) in page 2: do you consider in the system model some uncertainty in the measurements of the states x ? For example, it is said in the supplemental material that the velocity of the servo-motor of your second experiment is estimated using a high pass-filter, and is hence not perfectly known. If it is modeled, is it included in the process noise w or how do you deal with it ? This seems important as it is rather common not to have access to perfect measurements in controlled systems. Furthermore, in page 3 you restrict the study to static-gain policies. I believe that it would be useful for non-expert readers to comment on how restrictive this assumption is. You could say for example whether it is a common assumption and give one or two examples of systems which can be properly controlled that way. I also believe that the authors should comment the approximations made in section 4.1 leading to problem (7), on how the solutions of this convex surrogate problem are close or not to those of the original problem (3). Indeed, this seems to be an important element of the theoretical analysis here, as the method proposed allows to solve (7), while previous methods (DP) seem to solve a gridded version of problem (3). <--- I apologize for this comment. I reckon after reading the authors rebuttal that this question was indeed treated in the manuscript and that I just did not see it. ---> Moreover, no comparisons are made what so ever between the proposed approach and the classical Dynamic Programming approach (or other known Optimal Control methods). For example, in page 7, a complexity analysis of the RRL method proposed is given, but it is never compared to the complexity of DP. This seems crucial as the authors justify RRL in page 3 by the fact that DP is "computationally intractable for systems of even modest dimension ». As far as I’m aware, semidefinite programming problem also suffers heavily from dimensionality, which can indeed be seen in the complexity presented in page 7. In the same direction, experimental results in section 5 never compare RRL to SoTA methods. Also, how much time do the computations take ? Again, the motivation for not solving (3) with a known method seems to be that known methods are not efficient/tractable, and yet no results points toward the conclusion that the proposed method is efficient/practical in higher dimension. Also concerning experimental results: 1. In figure 2a, what happened in the experiments corresponding to the outliers of the RRL results plot? These seem to be worth investigating/commenting as the RRL method is supposed to deliver robust solutions. 2. In figure 2c, shouldn’t greedy method interpolate between RRL and Nominal, as it is first equal to Nominal than shifted towards RRL ? 3. More experiments with more real-word data would make the paper way more convincing. All in all, the ideas in this submission seem quite interesting and it could likely be significant for other researchers in the field, but this is somehow mitigated by the lack of experimental and theoretical comparisons to SoTA methods supporting the claimed strength of the new approach proposed. ----- I have read the authors rebuttal and I am mostly satisfied by their clarifications. I am hence increasing my score under the condition that they add indeed the new experiments and complexity analysis. PS: I don't understand however why the execution times that the authors are willing to add to their V2 was not included in the rebuttal.

Reviewer 2



Originality: Convex approximations to LQR-like problems have been studied extensively in the literature, and the authors are up-front about this. However, the bound on the estimation error, the convex reformulation of the infinite horizon LQR problem, and the convex semi-definite program formulation are all novel. Quality: The paper is of extremely high quality, and provides thorough proofs and analysis of all claims. The authors could spend more time comparing their algorithm to other benchmarks tasks (if only to situate the algorithm with others in the space), but the provided analysis seems more than sufficient for publication. Clarity: The work is quite clear. I particularly enjoyed the comparisons demonstrating the higher information gleaned by greedy algorithms (yet concomitant lower performance) because of their failure to automatically balance exploration/exploitation. Significance: While I am not overwhelmingly familiar with the space, this work seems like an extremely general tool for use on LQR type problems. I would appreciate more discussion of how this method compares with others in the space (if only for this reviewer's edification).

Reviewer 3



Originality ----------- The paper seems to be significantly novel, and the related work section touches on many related areas and prior work. Admittedly, I am no expert on the field of robust control, however my impression is that the authors have done their due diligence in citing prior work and delineating their contributions from these works. Quality ------- The quality of the work is perhaps my greatest concern. As I have mentioned, experiments were only conducted on numerical simulations and a simple hardware setup, and though the analysis on these two domains are decently thorough, it is rather unclear from just these domains how well the method scales or whether it provides tangible benefits on systems of actual interest. I would request that the authors do two things: (1) Elaborate further on the real hardware experiment. Is there a goal for controlling this system, i.e., why is this system of interest beyond demonstrating the capabilities of the method? The physical servo is connected to a synthetic linear system, which seems to make the experiment more toy as this limits the realisticness of the setup. Finally, is there any reason this system in particular benefits from robust control, in that there may be danger or costs associated with running the system? (2) Conduct further experiments, ideally on real robotic systems. As the authors do not assume that the underlying dynamics are known, it seems feasible to run this algorithm on, e.g., robot arms or locomotion platforms where the true dynamics are not linear. These systems seem much closer to reality and can benefit much more convincingly from robust control, as safe control is certainly desirable. If real robots are not an option, additional simulated systems could be useful, even extremely simple domains such as a 2D point mass navigation domain where the system truly is linear quadratic. This would have the benefit that the domain can potentially be engineered with obstacles, traps, etc., that make robust control extremely desirable. Clarity ------- In general, the paper is well-written except for some minor issues. Most egregiously, there is no conclusion, which makes it hard for readers to understand the key takeaways and limitations of the work. A couple of minor nits: - line 52, “essential for implementation on physical systems”: I would argue that this is only true for some physical systems - proposition 2.1: it may be useful to explain that the “uniform prior over the parameters” is an improper prior, or otherwise remove the need for a degenerate prior altogether from the analysis Significance ------------ As I have already discussed most of my concerns, I will simply state that the aforementioned improvements to the quality and clarity of the work would greatly improve the significance as well.

[Author Response · NeurIPS 2019]

Foremost, we would like to thank the reviewers and (S)ACs for giving up their time to conduct and organize the
reviews of our manuscript. Indeed, all reviewers have read the paper thoroughly, and we are grateful for their positive
comments and constructive feedback. Moreover, all three reviewers were extremely consistent in their suggestions for
improvement: i) include comparisons with SoTA methods, and ii) consider additional experiments.

**Comparison with SoTA** We now compare the proposed approach with two additional SoTA methods: i) `ad-rbst`:
the robust, adaptive method from *Dean et al. Regret Bounds for Robust Adaptive Control of the Linear Quadratic*
*Regulator, NeurIPS 2018*, and ii) `ts`: Thompson sampling. Results are presented in Fig a. Full details will be provided
in the revised manuscript (V2), but for now: our proposed method, `rrl`, performs better than the other robust method
`ad-rbst`. `rrl` also performs best in terms of worst-case cost (which it was designed to optimize), although `ts` achieves
the lowest cost on the true system, **when the policy is stable**. Note that the worst-cost for `ts` is infinite (unstable) in
99%, 54%, and 8% of trials at epochs one, two and three respectively. We also applied Thompson sampling to the
hardware-in-the-loop (HIL) experiment, and observed that it was unable to stabilize the system, cf. Fig b.

(a) Simulated example: worst-case (L), true system (C), information (R).    (b) Instability in HIL.    (c) Simulated pendulum.

**Experiments** Please allow us to first justify the use of the HIL experiment. The motivation for use of the servomecha-
nism was twofold: i) control of servomechanisms is a ubiquitous task in practice (e.g. robotics), ii) more importantly,
unlike e.g. the inverted pendulum, which is nonlinear, the planar servo can be reasonably well modeled (globally) as
a linear system. We believe that the servo represents a good compromise between the complexities of a real world
system (subtle nonlinearities such as backlash and friction, unmodeled dynamics, disturbances, etc) and a system
that approximately satisfies the assumptions of the method. Furthermore, with $n_x = 7$ & $n_u = 2$, the HIL system
is not of trivial dimension; many real-world systems can be represented with models of this size. Nevertheless, we
also applied the method to a (simulated) inverted pendulum (modeling the Quanser pendulum); cf. Fig c. Although
reasonable performance is observed, the results are not as 'clean' or 'explainable' due to the more complex behavior of
the nonlinear system (which does not satisfy the assumptions of the method). At the recommendation of the reviewers,
we can include these results in the paper (including experiments on the real physical pendulum), however, we believe
the HIL experiment is more compelling, as it is more consistent with the assumptions of the method.

**Reviewer #1** We wish to thank R#1 for identifying a number of areas in which the clarity of the manuscript could
be improved. All of the following points will be clarified in the revised manuscript (V2). ***Re: measurability of the***
***states,*** as in a number of recent papers (e.g. [11,12,13,16]), we assume that the states are directly measurable. Output
feedback control is indeed an important and common scenario, but this is beyond the scope of the paper. In the
example, we assume that the filter provides a suitable approximation of the velocity, and that any discrepancies can be
partially accounted for by the process noise. ***Re: static-gain policies,*** such policies are popular in practice, due to the
simplicity of synthesis and implementation; for instance, the (ubiquitous) proportional-derivative (PD) controller can be
implemented with static-gain, and by introducing an 'artificial state' (with known dynamics) to the system, integral
action (for PID) can be incorporated too. ***Re: comparisons to DP,*** we will add a brief discussion on the challenges of
'gridding' continuous state/action spaces in order to apply DP-based methods, citing relevant literature. As suggested,
we will report computation times in V2. ***Re: the approximations in §4.1,*** we attempted to discuss each approximation
in lines 150-156 and 159-160. However, as you rightly point out, given the importance of such assumptions, they will
be discussed in greater detail in V2. ***Re: the outliers in Fig 2a,*** This is an interesting question. Note that while all
methods were 'robust', sometimes the worst-case cost is quite bad (depending on the data realization). ***Re: Fig 2c and***
***interpolation,*** it is true that greedy first applies the 'nominal' controller; however, it also adds exploration 'noise' such
that greedy and RRL have the same worst-case cost at the first epoch. We must apologize for a small error: the center
figure in 2c is actually the cost on the true system. This is why the cost of greedy and RRL differ at the first epoch.

**Reviewer #2** Thank you for your positive comments and useful suggestions; please cf. '§SoTA' and '§Experiments'.

**Reviewer #3** We wish to thank R#3 for the many useful suggestions to improve the manuscript. ***Re: the conclusion,***
we apologize; this was simply due to space restrictions. We will endeavor to include a conclusion in V2. ***Re: the use***
***of the word 'essential',*** we completely agree; we will amend this (to 'certain physical systems'). ***Re: the uniform***
***prior,*** this is an excellent suggestion; we will explain that the prior is degenerate. ***Re: the hardware experiment,*** the
synthetic system was introduced to increase the dimensionality of the system so as to make the set-up somewhat
more interesting/realistic. ***Re: robustness of the HIL experiment,*** please cf. Fig b for consequences of instability.
***Re: further numerical experiments,*** for real-hardware experiments, please see above. Concerning more complicated
simulations, e.g. navigating obstacles, this is a very interesting suggestion. We feel that this goes beyond the scope of
the present paper, as it would necessitate higher-level path planning, but is an interesting direction for future work.

[Meta-Review · NeurIPS 2019]

The paper presents a new technique for robust optimization and balanced exploration in LQR problems. The technique is quite innovative since it leverages semidefinite programming instead of dynamic programming. This is an important algorithmic contribution with solid theory. For the empirical evaluation, the authors are expected to include the new experiments and running times mentioned in the rebuttal. Overall, this is very nice work.